# Tactile Sim-to-Real Policy Transfer via Real-to-Sim Image Translation

**Alex Church**
Department of Engineering Mathematics
Bristol Robotics Laboratory
University of Bristol, U.K.
`ac14293@bristol.ac.uk`

**John Lloyd**
Department of Engineering Mathematics
Bristol Robotics Laboratory
University of Bristol, U.K.
`jl15313@bristol.ac.uk`

**Raia Hadsell**
Google Deepmind
U.K.
`raia@google.com`

**Nathan F. Lepora**
Department of Engineering Mathematics
Bristol Robotics Laboratory
University of Bristol, U.K.
`n.lepora@bristol.ac.uk`

**Abstract:** Simulation has recently become key for deep reinforcement learning to safely and efficiently acquire general and complex control policies from visual and proprioceptive inputs. Tactile information is not usually considered despite its direct relation to environment interaction. In this work, we present a suite of simulated environments tailored towards tactile robotics and reinforcement learning. A simple and fast method of simulating optical tactile sensors is provided, where high-resolution contact geometry is represented as depth images. Proximal Policy Optimisation (PPO) is used to learn successful policies across all considered tasks. A data-driven approach enables translation of the current state of a real tactile sensor to corresponding simulated depth images. This policy is implemented within a real-time control loop on a physical robot to demonstrate zero-shot sim-to-real policy transfer on several physically-interactive tasks requiring a sense of touch.

Video results: *https://sites.google.com/my.bristol.ac.uk/tactile-gym-sim2real/home*.
Code: *https://github.com/ac-93/tactile_gym*.

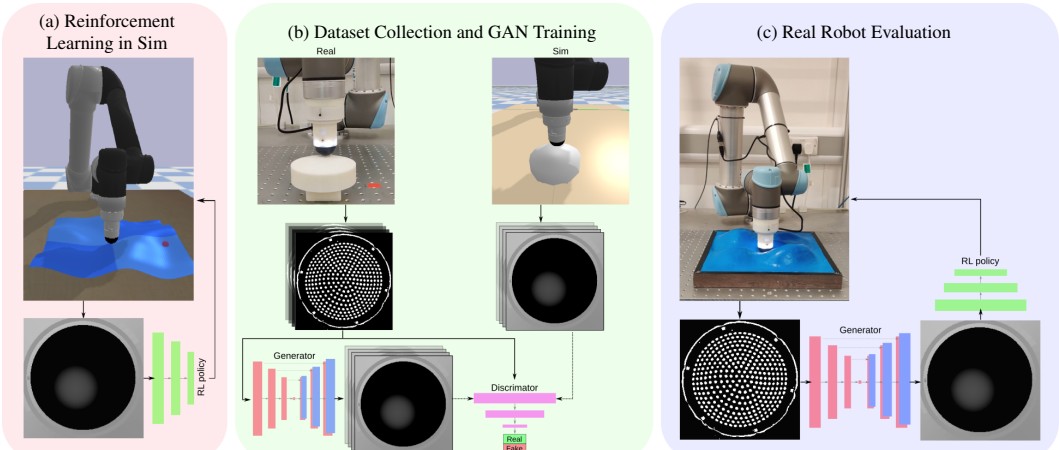

Figure 1: Overview of the proposed approach for sim-to-real transfer of learned tactile policies. a) Learn a policy in simulation directly from simulated tactile images. b) Train a GAN for translating between real and sim images using a dataset of image pairs collected by sampling a static environment with features similar to the environment used for reinforcement learning. c) Evaluate on the real robot by passing real images through the generator and then generated images through the RL policy.

5th Conference on Robot Learning (CoRL 2021), London, UK.

# 1 Introduction

Learning algorithms have an innate appeal for robotics to enable general, complex behaviours that would be difficult to achieve with classic control methods. Large-scale data is often required for robot learning, and so physics-based simulation has a vital role to play for data collection. A common approach in learning-based robotics is to simulate data that would be impractical to collect in reality, learn control policies from this data, and then transfer learned skills to the physical system. Simulation also offers advantages such as avoiding damage during exploratory training, exploiting privileged information from the simulator, and use of open-sourcing for dissemination of research findings. However, physics engines necessarily approximate the real world to reduce computational costs, giving a 'sim-to-real gap' that impairs the performance of these policies applied to reality.

Reinforcement Learning (RL) robotics research is dominated by the use of proprioception and vision as the sensory inputs. Whilst this can lead to complex behaviours [1, 2, 3], other important sources of information have been underutilised. Specifically, humans use the sense of touch to accomplish complex manual tasks, utilising a granularity of detail unavailable to our other senses. Tactile data has several advantages as the main source of information for learning: it does not suffer from occlusion, particularly for fine manipulation; contact information can be more detailed than visual images of an entire scene; and the observation space is constrained (e.g. to tactile images of markers), simplifying the translation of real to simulated data and, likewise, from simulated to real policies.

Here we make the following contributions aimed at bringing together reinforcement learning and tactile robotics: **1)** We provide an open-source suite of RL environments tailored to tactile robotics, utilising a simple and fast method of simulating a tactile sensor. **2)** We show that an RL method, Proximal Policy Optimisation (PPO) [4], acquires successful policies for all of these tasks. **3)** We demonstrate and validate a data-driven approach for zero-shot sim-to-real policy transfer via image translation between real and simulated tactile images on these tasks.

# 2 Related Work

**Sim-to-Real Transfer:** For computational efficiency, physics engines necessarily approximate the real-world dynamics, leading to a sim-to-real gap where dynamics and visuals differ between simulation and reality. This gap can make it difficult to transfer skills learned in simulation to the real world. Several methods have been proposed to mitigate this issue, namely *Domain Randomisation*, *Network Distillation* and *Domain Adaption* [3].

In this work we mainly use *Domain Adaption*, following James et al. [5] to train an image-conditioned generator network that translates between real and simulated images. When using vision, *Domain Randomisation* was necessary on simulated images to train a generator robust enough to generalise to real visual images. Here we exploit the more constrained image space of optical tactile sensors to train generator networks using a controlled tactile dataset. We perform some randomisations to make simulated tasks more difficult and RL policies more robust, aiming to keep within realistic conditions. Although privileged information from the simulator is used to construct a reward for improved learning, this information is not part of the observation in any task. As we perform zero-shot policy transfer from sim-to-real, *Network Distillation* is not required.

**Tactile Sim-to-Real:** Recently, several works have aimed to bridge between tactile simulation and reality. These split into two categories: simulating the physics of sensor deformation via Finite Element (FE) methods or replicating the captured sensor information via image rendering techniques. A concise overview of the sim-to-real approach applied to non-vision based sensors is given in [6].

FE methods have seen recent advancements in computational efficiency and accuracy. Narang et al. [7] simulated deformations of the BioTac sensor that, in combination with data-driven learning, can accurately regress the sensor output. Later, a 75-fold improvement in simulation speed was achieved with GPU-acceleration [6], with each simulated sensor output taking up to 5 seconds which is still impractical for sample-inefficient methods such as DRL. A similar approach was applied to a marker-based optical tactile sensor for regressing contact force fields [8, 9]. Later work [10] achieved zero-shot sim-to-real transfer of a complex policy learned in simulation via RL; however, task-specific approximation was needed for computational efficiency, e.g. assuming planar motion.

Most comparable to our work, Ding et al. [11] use an elastic deformation approach on the same tactile sensor as used here (a TacTip soft biomimetic optical tactile sensor [12, 13]). They focused on supervised learning from simulated marker positions rather than tactile images, and with domain randomisation could accurately predict edge position and orientation for use in contour following. The authors noted that the simulation speed is restricted by Unity's collision detection module.

A computationally efficient approach is to render images from specific optical tactile sensors with rigid-body physics approximating the contact dynamics. Depth images from a GelSight tactile sensor were rendered using the Gazebo physics engine with additional Gaussian smoothing, light rendering and image calibration to help match real tactile images [14]. On a supervised classification task, a drop of 39% accuracy was found for direct sim-to-real transfer, reduced to 6.5% with texture augmentation as a form of domain randomisation. A similar approach is proposed in [15] with extended image-rendering techniques applicable to a range of GelSight-type sensors on more complex physically-interactive tasks; however, so far no sim-to-real quantitative results have been reported.

Our work here focuses on a high-resolution tactile simulation environment appropriate for RL of complex control policies. By relying on rigid-body approximations to the contact dynamics, our physics simulation is significantly faster than FE methods without task-specific approximations. Our simulated TacTip images use depth image-rendering techniques like those used for the GelSight [14, 15]. However, a key novelty is that our simulated depth images do not closely resemble the real TacTip images, because depth is non-trivially related to marker shear for the TacTip (and closely related to image shading for the GelSight). Instead, we close the sim-to-real gap by relying on a later domain adaption phase. Hence, we keep the simulation agnostic to the sensor transduction, which should facilitate future application of these methods to other types of tactile sensor.

## 3  Tactile Simulation

In this work, we utilise PyBullet's synthetic camera rendering to capture depth images within a virtual optical tactile sensor, based on the CAD files used to 3D print a real TacTip (see e.g. [12, Fig. 4]). When gathering tactile images, we take the difference between the current depth image and a reference depth image taken from when the sensor is not in contact. This difference produces a penetration depth map that generalises to arbitrary sensor shapes. Noise is removed from the image by zeroing values below a set tolerance of $10^{-4}$, followed by a re-scaling from $[0, \text{max\_penetration}]$ to $[0, 255]$. An artificial border is also overlaid onto the tactile image to bring it closer to real tactile images and to provide a reference point that transforms with augmentations.

In order to achieve the computational efficiency necessary to generate data at large scales, we approximate the soft tip of the tactile sensor with rigid body physics simulation. We limit the contact stiffness and damping used during collision detection as this allows penetration of objects into the simulated sensor tip in an approximate manner to the deformation of the real tip.

We describe our simulation method in relation to the list of desiderata proposed by Wang et al. [15].

**High Throughput:** We use PyBullet's GPU rendering functionality to offer fast simulation. On a PC with an Nvidia 2080Ti, we can achieve up to $1000$ fps when rendering single tactile images at $128 \times 128$-resolution. Multiple PyBullet physics engines can also run in parallel, so during training we used 10 vectorised environments to increase throughput.

**Flexible:** Whilst this study is predominantly based on the TacTip sensor, we do not attempt to simulate images accurate to any specific tactile sensor. Instead, we simulate only useful tactile features, relying on a later image-translation stage to map from real to simulated images. We expect similar results are possible with a broad range of other high-resolution optical tactile sensors, including sensors of the Gelsight type [16, 17, 18], the Soft-Bubble [19] and optical shear-based sensors [20].

**Realistic:** Instead of aiming to realistically simulate the physical properties of any specific sensor, which is both difficult and computationally expensive, we simulate only the desired properties of an idealised tactile sensor, currently focussing only on contact geometry. Our data-driven approach helps bridge the sim-to-real gap, so there is no need to generate synthetic images that match images from a real sensor to high precision.

**Ease of use:** The simulation suite has been open-sourced. The simulation only requires the commonly-used PyBullet, so our approach should have a low barrier of entry. Our approach should also extend readily to different tasks and other tactile sensors.

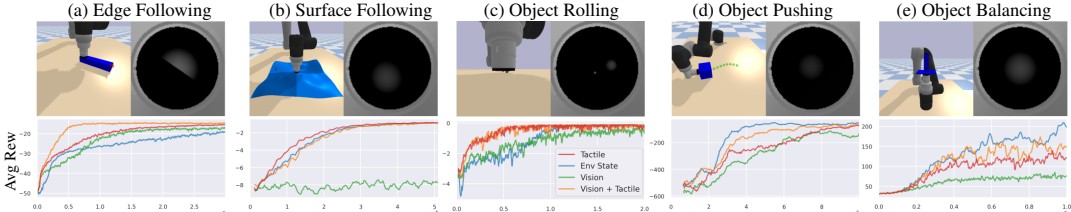

Figure 2: The DRL method PPO learns successful policies for the five considered RL environments. (a) Traversing a randomly oriented edge while maintaining a set pose. (b) Traversing a randomly-generated 3D surface while maintaining a set penetration and normal orientation. (c) Manipulating a ball from a random initial position to a goal location. (d) Manipulating a cube along a randomly-generated trajectory. (e) Stabilising an object on the sensor tip. Panels show mean reward during training, smoothed with a window size of 50 and averaged over 3 seeds. Task reward commonly consists of negative distances between target and current poses, further details in Appendix C.

# 4 Reinforcement Learning Environments

Each task is provided with a set of observation spaces to allow for verification of the environments, comparisons between tasks, and an examination of multi-modal visuotactile control. Four standard types of observation space are considered:

**Env State:** Comprises state information from the simulator, which is difficult information to collect in the real world. We use this to give baseline performance for a task that is expected to act as an upper limit. The information in this state varies between environments but commonly includes tool center point (TCP) pose, TCP velocity, goal locations and the current state of the environment.

**Tactile:** Comprises images ($128 \times 128$) retrieved from the simulated optical tactile sensor attached to the end effector of the robot arm (Figure 2 right). Where tactile information alone is not sufficient to solve a task, this observation can be extended with state information retrieved from the simulator. This can only include information that could be easily and accurately captured in the real world, such as the TCP pose that is available on industrial robotic arms and the goal pose.

**Visual:** Comprises RGB images ($128 \times 128$) retrieved from a static, simulated camera viewing the environment (Figure 2 left). Only a single camera was used, although this could be extended to multiple cameras. As the simulated environment differs visually from the real-world environment, sim-to-real using RGB observations is challenging, requiring an approach like that of [5, 21].

**Visual + Tactile:** Combines the RGB visual and tactile image observations to into a 4-channel RGBT image. This case demonstrates a simple method of multi-modal sensing.

## 4.1 Tactile Exploration Environments

In this set of tasks, the robot interacts with a physical environment that stays static, with the objective to learn policies to safely explore physical areas. These policies can be used to report information about novel objects, simplify human-control methods and improve operational safety of the robots. Two exploration environments have been created, for edge and surface following. More complete details on these environments are given in Tables 5 and 6 of Appendix C.

**Edge Following:** The edge-following environment is used to train an agent that maintains its pose relative to an edge whilst traversing the edge towards a goal location. Previous work has demonstrated that robust 2D contour following can be achieved with supervised learning techniques [22]. Here we demonstrate this task can also be completed via sim-to-real reinforcement learning.

**Surface Following:** The surface-following environment is used to train an agent that maintains a set contact penetration of the sensor whilst orientating the TCP normal to an undulating 3D surface generated using OpenSimplex noise [23]. For ease of use, we automatically direct the sensor in the direction of the goal, analogous to the pose-based tactile servo control methods introduced in [24].

## 4.2 Non-prehensile Manipulation Environments

In this set of tasks, the objective is to learn policies that manipulate external objects in a desired manner. As our focus is on single sensors in this work, we consider non-prehensile manipulation. However, the simulation does support rendering of multiple tactile images, which could in principle

be used simulate tactile grippers, manipulators or multiple robot arms in future work. Three manipulation tasks are proposed: object rolling, object pushing and object balancing. More details on the non-prehensile manipulation environments are given in Tables 7, 8 and 9 of Appendix C.

**Object Rolling:** The object-rolling environment requires the manipulation of small spherical objects into a goal position within the TCP coordinate frame. The agent must learn a policy to roll the object from a random starting position to a random goal position. In this environment, a flat tactile sensor tip is used to simplify the motion needed to maintain contact with a spherical object.

**Object Pushing:** In this task, the objective is to push an object to a goal location using the tactile image, as considered in previous work using supervised learning techniques [25]. Here we consider a cube pushed along a randomly-generated trajectory. The initial pose of the cube is randomised within limits and the trajectory generated using OpenSimplex noise.

**Object Balancing:** This task is analogous to the well-known 2D inverted pendulum problem [26], where an unstable pole with flat base is balanced on the tip of a sensor that points upwards. A random force perturbation is then applied to the object to cause instability. The objective is to learn a policy that applies planar actions to counteract the rotation of the balanced pole.

### 4.3 Reinforcement Learning Results

This work uses the Stable-Baselines-3 [27] implementation of PPO to train all tasks (further details in Appendix B). For an initial comparison, we train all tasks using all available observation spaces, averaged over 3 seeds. Training results are given in Figure 2, deterministic evaluation results and more detailed plots are given in Appendix D.

Environment state observations are used to verify that the environment leads to desired policies. As no image rendering is performed, this runs faster than tactile or visual observations; hence, it was used to find stable hyperparameters. Although comprised of ideal state information, we find that tactile image observations can lead to more sample-efficient and stable training, which is most clearly visible in the Edge-Following and Object-Rolling environments (Figs 2a, 2c).

Policies trained exclusively using visual observations tended to perform worst, which we attribute to most tasks being targeted towards tactile data; for example, details of the contact can be obscured using a single external camera. The best visual agent performance was for the object pushing task, which is the coarsest manipulation challenge. Whilst the object rolling task appears to perform well with visual observations, there is a notable performance gap where visual polices do not accurately learn the desired behaviour. When combining visual and tactile observations, successful learning can take place. Despite the additional complexity in the observations, agents can learn well with notable improvements in sample efficiency for the Edge-Follow and Object-Push tasks (Figs 2a, 2d).

## 5 Real-to-Sim Image Translation

An advantage of using tactile images as the main form of observation is that the image space is more simplistic (e.g. markers on a uniform background) compared with the variety of visual images from a scene. Vision-based images are affected by features such as changing lighting conditions, shadows and texture that usually constitute a superfluous level of detail when solving a task. Simulating these visual features can make the learning more difficult because of the increased complexity of the observations and the added computation to render these images; conversely, ignoring these visual features would make image translation more difficult because of the greater simulation and reality gap, as they are still prevalent in reality, necessitating techniques such as image randomisation or RL task-aware training [5, 21]. These complexities are not present for the real and simulated tactile images considered here because the camera is confined within the enclosure of the tactile sensor.

Conversely, a disadvantage of using tactile images as the main observations is that the tasks require the robot to be in close proximity, or touching, the environment. Whilst this contact may be necessary for some tasks, such as manipulation, it does make exploration in reality a challenge because of the self-inflicted damage that can occur. The approach proposed here is to use a separate data-collection stage, where the tactile sensor explores a series of configurations in a safe and controlled manner. As the tactile image space is relatively confined, an efficient exploration of a representative sample of configurations for a specific RL task becomes possible. Whilst some sensor configura-

tions are not possible to sample in reality, such as large penetrations of the sensor that would cause damage, we aim to train a model that generalises to those unreachable configurations.

Specifically, in this work we treat the sim-to-real problem as a supervised image-to-image translation problem. The same data collection procedure is performed in both simulation and reality, producing a data set of real and simulated image pairs. Here we choose to transfer from real-to-sim images, because the real images are richer in information with details such as shear forces that we choose not to model in simulation. Generative Adversarial Networks (GANs) are the state of the art for realistic image generation. Here we use the pix2pix [28] architecture for image-to-image translation, which uses the U-net [29] architecture for the image conditioned generator and a standard convolutional network for the discriminator (Appendix E shows this architecture applied to tactile images).

## 5.1 Data Collection

Three data sets corresponding to distinct environments are collected for sim-to-real transfer, each with 5000 image pairs for training and 2000 pairs for validation. We approximate the task environment with a simplified static environment consisting of 3d-printed components and no specialist hardware. Per task, these datasets take $\sim$5 hours to collect on physical hardware and $\sim$100 seconds in simulation. Following a previous investigation of contact-induced shear with the same optical tactile sensor [30], we deliberately induce shear perturbations by randomly sliding the sensor during data collection (details given in Appendix F). Past work has shown this step is key to ensuring that the trained neural network outputs are insensitive to unavoidable motion-dependent shear during task performance (for more details, see [24]). We do not model any sensor shear in simulation to ensure that the real-to-sim image generation is also insensitive to shear on the real sensor.

## 5.2 Pix2Pix Training

The only change to the pix2pix architecture was to replace instance normalisation with spectral normalisation, with other default parameters sufficient for training on tactile images (full details in Appendix E). This change reduced droplet artefacts that were otherwise prevalent in our generated images. In addition, excluding the border from the simulated tactile images improved training, as otherwise the GAN focused on generating a realistic border instead of the tactile imprint. This is important because the imprint is the useful component when learning control policies. Instead, the border is re-added from a saved reference image after generation occurs.

Accurate tactile image generation is achieved with minimal difference between generated and target images (example images for each data set shown in Appendix F, Figure 11). For the edge, surface and probe datasets, mean SSIM scores across the full validation set are 0.9955, 0.9924 and 0.9942 respectively. The source of the small errors are highlighted in the shown SSIM image differences (Figure 11, right). The imprint borders in the generated images lack some sharp details generated in simulation, which will likely be due to a slight elastic stretching of the skin of the real tip that is not modelled in simulation. An approach such as Gaussian smoothing of the simulated images, as in [14], could reduce this effect, although we did not find it necessary to explore in the present work.

Crucially, the generator can interpret and generate image features useful to training RL policies. For example, edge orientation and position are accurately captured in the GAN. Also, the generator can generalise to images unseen during training. For example, all training data only had imprints from one source in each training image; however, during inference, multiple sources can be applied and the generator still produces realistic outputs (video demonstrations available *here*). Similarly, the generator appears to generalise outside the training data for penetration depth, which is important for helping to ensure that the sensor can avoid damage by not pressing too far into a surface.

## 5.3 Supervised Learning Comparison

For comparison to the only existing sim-to-real approach for the TacTip [11], we implement a supervised learning task to predict the radial displacement and polar angle of an edge pressed into the sensor (results in Appendix G). Domain adaptation with our proposed method of tactile simulation out-performs the previous approach using elastic deformation simulation and domain randomisation, with approximately a 3-fold improvement in Mean Absolute Error (MAE) of the predictions.

| Real | Sim | Real | Sim |

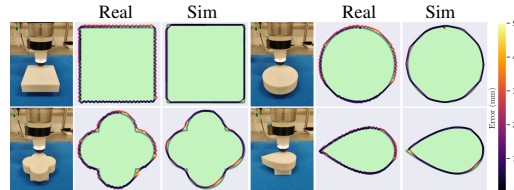

Figure 3: Comparison between sim and real performance for the Edge-Following RL environment. Using several flat shapes (circle, square, clover, foil) to evaluate the policy generalisation performance.

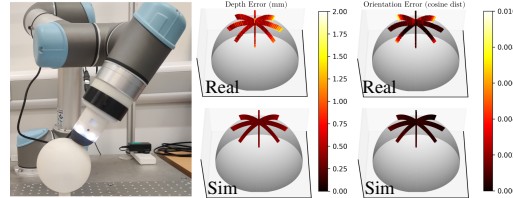

Figure 4: Comparison between sim and real performance for the Surface-Following RL environment. Positional error (left), Orientation error (right), Real data (top), simulated data (bottom).

# 6   Results: Sim-to-Real Policy Transfer

Successful sim-to-real policy transfer was achieved on 4 of the 5 proposed tasks. Qualitative demonstration videos are available at *https://sites.google.com/my.bristol.ac.uk/tactile-gym-sim2real/home*.

**Edge Following:** The learned edge-following behaviour is evaluated over several novel flat shapes in the environment (Figure 3). The learned policy successfully traverses all objects, generalising to novel features such as curved edges and sharp internal or external corners not present during training. As these shapes are 3D printed, we have access to the ground truth in the CAD files (accurate to

Table 1: Quantitative results for edge following task. Mean distance from ground truth shape taken over 1000 evaluation steps.

|  | Real | Sim |
| --- | --- | --- |
| **Square** | 1.47mm | 0.63mm |
| **Circle** | 1.50mm | 0.80mm |
| **Clover** | 1.58mm | 1.38mm |
| **Foil** | 1.09mm | 0.47mm |

the precision of the 3D printer). We compare the trajectories taken in simulation to those in reality, reporting the distance from each point of the trajectory to the nearest ground truth node (Table 1). The results show that the sensor maintains close proximity to the target edge.

The drop in performance on the physical robot seems related to an oscillating behaviour also present in simulation. The movements are exaggerated in reality due to increased latency between capturing an observation and predicting an action. A likely cause is the action-range clipping used in the PPO algorithm, which results in the action extremes being more prevalent. Approaches such as squashing functions [31], beta distributions [32] or constraint-based RL [33] could mitigate this artefact.

**Surface Following:** To evaluate the surface-following behaviour, a spherical object is traversed from its centre outwards in a set direction ranging over $360°$ in $45°$ intervals (Figure 4). As with the edge following task, we use a 3D-printed shape whose ground truth is known from the CAD model. The sensor successfully and accurately

Table 2: Quantitative results for surface following task. Mean distance from ground truth evaluated over 1000 steps.

|  | Real | Sim |
| --- | --- | --- |
| **Depth Error** | 0.57mm | 0.30mm |
| **Cosine Error** | 0.00118 | 0.00054 |

traverses the object on the physical robot despite texture and frictional forces not modelled in simulation. Like the edge following task, the accuracy drops between sim and real, although the evaluated policies still exhibit the desired behaviour. A more extensive qualitative test is given in the supplementary video for the undulating surface shown in Fig.1.

**Object Rolling:** The object-rolling behaviour is tested by manipulating ball bearings (on an extended dimension range of 2-8mm diameter) from random initial positions to random goal locations. During the evaluation, the position of the ball bearing relative to the sensor is obscured, so the imprint of the object is tracked on the tactile image using basic computer vision (blob detection). The target is reached when the pixel distance is less than a 5-pixel threshold ($\sim$2mm). On the physical robot, this is performed for 25 trajectories with each bearing, resulting in 100 consecutive successful trials (subset shown in Figure 5). To compare with simulated results, we apply the same initial and target positions then perform the same pixel tracking. The sim and real results are similar, albeit with greater noise in the real trajectories.

**Object Pushing:** To evaluate this task we tracked the objects following the method given in [25]. Importantly Aruco markers were only used for obtaining quantitative results and not as part of the observation. This task was the most challenging to achieve accurate perfor-

Table 3: Quantitative results for object pushing task. Mean Euclidean distance from trajectory.

|  | Straight | Curve | Sine |
| --- | --- | --- | --- |
| **Real Cube** | 11.5mm | 11.7mm | 13.6mm |
| **Sim Cube** | 11.1mm | 10.1mm | 12.7mm |
| **Real Tri** | 11.0mm | 14.0mm | 12.6mm |
| **Sim Tri** | 21.8mm | 10.2mm | 11.0mm |
| **Real Cyl** | 13.3mm | 16.7mm | 9.9mm |
| **Sim Cyl** | 24.1mm | 13.9mm | 12.7mm |

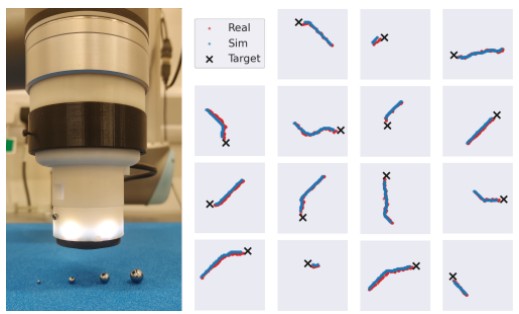
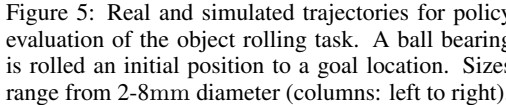
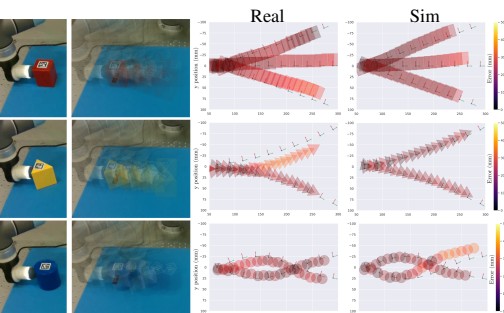

Figure 5: Real and simulated trajectories for policy evaluation of the object rolling task. A ball bearing is rolled an initial position to a goal location. Sizes range from 2-8mm diameter (columns: left to right).

Figure 6: Evaluation of the pushing task. Trajectories show a cube (top), triangular prism (middle) and cylinder (bottom) pushed along straight, curved and sinusoidal trajectories respectively.

mance, likely due to the approximations of the simulated sensor tip, object being manipulated, and frictional interactions. Despite this, when evaluated in reality, task success was achieved across several trajectories with objects of different shape to that seen during training (examples in Figure 6). Performance both in simulation and reality was sensitive to initial conditions and perturbations, with compounding errors causing the object to veer off the trajectory. Future work could improve robustness by learning under additional dynamics randomisation, introduction of random force perturbations or by better matching the physical properties of the real objects.

**Object Balancing:** The physical task remains an outstanding challenge, because the successful training in simulation required physics parameters outside the capabilities of our UR5 robot.

# 7 Discussion and Future Work

In this work, we demonstrated that zero-shot sim-to-real policy transfer is a viable approach for tactile-based RL agents. To learn the policy, we created a fast method for simulating tactile images based on contact geometry. From these images, distinct policies were learned for several physically-interactive tasks requiring a sense of touch. We demonstrated that a data-driven model for real-to-sim image translation can be embedded into the control loop for successful sim-to-real policy transfer. There are several future directions for extending and improving this approach.

The tactile simulation of both the contact dynamics and captured information in the model could be improved. Our contact dynamics model used used rigid bodies with soft contacts as an approximation to the deformation of real tactile sensors. Extending this work with soft-body deformation should enable a more realistic simulation, which was not pursued in this initial study because of increased computational costs and instability issues with the simulators. The captured information in the present study focused exclusively on contact geometry, which in principle could be extended to include global shear information through contact data available in most physics simulators. Texture could be also included using rendering techniques common in photo-realistic image generation. Local shear of the sensor would be challenging to simulate, for example during incipient slip, due to the contact reduction commonly used in physics engines for computational efficiency; however, this capability may be possible in soft-body simulations where these local forces are required.

For translating real to simulated images, improvement could be made to both the GAN and training data. We used a conventional pix2pix [28] approach, which can be improved with extended methods [34]. In addition, a distinct data set was collected for training the GAN in each learning environment. Because of the constrained nature of the tactile image space, a general tactile data set could potentially be used train a single image-translation model for policy transfer over multiple tasks.

The environments considered here focused on skills achievable with only a single sensor and where the optimal behaviour is obvious for a human to interpret, as needed to verify that RL with a sim-to-real approach is a viable method. Future work could focus on tasks where ideal control policies are more complicated or unknown, so the RL framework can be fully exploited. An interesting topic would be to extend these methods to prehensile tasks such as grasping and dexterous manipulation, where tactile information will be valuable for learning desirable and robust policies. Although only one sensor was used in this current work, our tactile simulation does support multiple sensors, offering the opportunity to extend to more complex tasks involving hands with multiple tactile fingertips.

## Acknowledgements

Alex Church was supported by an EPSRC CASE award sponsored by Google DeepMind. John Lloyd and Nathan Lepora were supported by a Research Leadership Award from the Leverhulme Trust on 'A biomimetic forebrain for robot touch' (RL-2016-39).

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
