# OpenReview forum: "Tactile Sim-to-Real Policy Transfer via Real-to-Sim Image Translation"
_robot-learning.org/CoRL/2021/Conference — CoRL2021 Poster_

### Official Review · Reviewer_Cj9a · 2021-07-21

**Originality:** Very Good
**Technical Quality:** Very Good
**Clarity Of Presentation:** Excellent
**Impact:** 4

**Recommendation:**

Strong Accept: I recommend accepting the paper and will argue for my recommendation even if other reviewers hold a different opinion.

**Summary:**

This paper applies sim2real RL policies for several real-world robotic tasks that benefit from tactile sensing. The authors show that PPO can successfully solve the tactile tasks in simulation. To bridge the sim2real gap for policy transfer, the authors apply real2sim using GAN, which transforms complex marker arrays to simulated contact depth. Evaluation is done in both simulation and real environments. The authors plan to open-source the tactile simulator and environment for TacTip, which can benefit future research.

**Issues:**

As stated in the Weaknesses:
- For the proposed tasks, feeding a low-dimensional contact state (like contact center position and major contact orientation) combined with simple PID controllers seems to be sufficient enough to solve the tasks. The whole pipeline of RL and GAN seems to be solving simplified problems.
- Among the tasks, the pushing task is the most challenging and interesting one, since it has to deal with objects with unknown dynamics. But there is no clear evidence that the proposed method is better than any baseline or intelligent/robust to uncertainties in the real world scenario. It is suggested to add some disturbance to study how robust the system is.

**Reviewer Expertise:**

Very good: Comprehensive knowledge of the area

**Strengths And Weaknesses:**

Strengths:

- The paper is clear and easy to follow.
- It is novel to apply GAN to bridge the real2sim gap for tactile signals. The authors put extra consideration to disentangle the influence of shear force for better real2sim.
- It is appreciated for evaluating the RL policy for both simulated and real environments and comparing the performance of different modalities in simulation.
- Supplementary video demonstrates well the performance of the system in simulation & real environments.
- The codes are planned for open-sourcing.

Weaknesses:

- For the proposed tasks, feeding a low-dimensional contact state (like contact center position and major contact orientation) combined with simple PID controllers seems to be sufficient enough to solve the tasks. The whole pipeline of RL and GAN seems to be solving simplified problems.
- Among the tasks, the pushing task is the most challenging and interesting one, since it has to deal with objects with unknown dynamics. But there is no clear evidence that the proposed method is better than any baseline or intelligent/robust to uncertainties in the real world scenario. It is suggested to add some disturbance to study how robust the system is.


**Summary Of Recommendation:**

The authors put good effort to push towards benchmark and sim2real policy transfer for tactile robotic tasks. It’s novel to apply GAN for tactile signals to bridge the real2sim gap, and the framework can potentially benefit many future works. Evaluations on simulation and real-world are appreciated. It can be further improved with some more challenging tasks, comparing the system with baseline control policies in the real world, and studying how intelligent/robust the system is under disturbances.

---

> ### Author Response · Authors · 2021-08-27
> **Response to Reviewer-Cj9a**
>
> We thank the reviewer for their endorsement and insightful comments.
>
> **Strengths:**
>
> Thank you for recognizing the value of these contributions of this work.
>
> **Weaknesses:**
>
> Pipeline of RL and GAN seems to be solving simplified problems:
>
> In this work we chose to use RL for its general applicability and potential to scale with observation and task complexity. Additionally, we found other advantages to the choice of RL such as having a single general method that can be applied across several tasks with minimal change simplifying implementation and learning from scratch inputting less bias into the system.
>
> In our opinion, it made sense to verify that an approach will work initially on relatively simple problems (in comparison with e.g. dexterous manipulation). With that said, each task does contain some complexities and we wouldn’t describe them as simple.
>
> Sufficiency of low-dimensional contact state:
>
> This is an interesting point, we agree that a low dimensional contact state will be able to solve some of these tasks, particularly in simulation. With this approach, there remains the issue of transferring to the real domain, which will likely require either a) a method to predict contact position/force/normal through calibration with a force torque sensor or b) a method to predict the contact state given by simulation. For method b) an approach similar to what we presented here could be applicable, although modified to work with lower dimensional data in place of images. As more information is captured with contact geometry (which implicitly contains contact location and force), we deemed it more practical to directly learn this mapping.
>
> Disturbance study on pushing task:
>
> We agree that of tasks evaluated on the real robot the pushing task was the most challenging. As mentioned in the paper the results were sensitive to initial conditions, we additionally mention perturbations in the revised version. Additionally, we give some suggestions of future methods that could improve robustness to perturbations and sensitivity to initial conditions (“Future work could improve robustness by learning under additional dynamics randomisation, introduction of random force perturbations or by better matching the physical properties of the real objects.”).

---

> > ### Comment · Reviewer_Cj9a · 2021-09-03
> > **Post-Rebuttal Comment**
> >
> > I thank the authors for their response.
> > The responses make sense. Looking forward to future works for extending the method for various complex manipulation tasks!

---

### Official Review · Reviewer_3knc · 2021-07-23

**Originality:** Good
**Technical Quality:** Very Good
**Clarity Of Presentation:** Very Good
**Impact:** 3

**Recommendation:**

Weak Accept: I recommend accepting the paper, but will not argue for my recommendation if the majority of other reviewers have a different opinion.

**Summary:**

In this work, the authors propose a method for successful sim-to-real transfer of manipulation policies utilizing tactile perception for a common type of optical tactile sensors. More specifically the authors present:
- A simple and fast simulation method for tactile sensors, where contact geometry is represented as depth images;
- Data driven algorithm to translate current state of a real tactile sensor to state of tactile sensors in simulation.
- Successful results of learning various manipulation policies utilizing tactile perception in simulation.
- Successful zero-shot sim-to-real transferability of the learned policies in majority of tasks.


**Issues:**

The authors mentioned a possible cause of oscillations and several potential solutions to eliminate action range clipping in the policies. It would be interesting to have an investigation of these effects, as well as more extensive investigation of generalization capabilities of the approach.

**Reviewer Expertise:**

Good: General knowledge of the area

**Strengths And Weaknesses:**

Strengths:
- The paper is well organized. It is interesting to read and simple to follow.
- The paper tackles problems on the intersection of two important topics: tactile perception and sim-to-real transfer.
- The paper proposes a method that can find practical applications in various real world settings. Successful results strongly support the applicability of the approach.

Weaknesses:
- It seems that the method would work well with just certain types (albeit a quite popular set) of tactile sensors. It will likely fail with more complex sensors, such as BioTac. Overall, it is important to demonstrate what specific sensors it can and cannot work to see the limitations of the approach.
- The method requires a separate procedure to learn real-to-sim mapping of the sensory data models. It is a bit hard to grasp from the description in the paper how complicated is the setup for such data collection?
- As far as I understood, the paper does not strongly demonstrate sim-to-real generalization toward strong variation of the tasks themselves. Can it follow new contours and edge shapes? Push various types of objects? Roll other objects (at least various sizes). It does seem like the simulated scene of the setup should match parameters of the actual experiment, which may, in some cases, limit applicability of the approach.


**Summary Of Recommendation:**

Despite moderate scientific contribution and not overwhelming scope, this is a well written and interesting work that can find its applications. I am recommending “weak accept”.

---

> ### Author Response · Authors · 2021-08-27
> **Response to Reviewer-3knc**
>
> We thank the reviewer for their constructive comments and recognition of its value.
>
> **Strengths:**
>
> We are pleased to hear you find the work interesting and recognize the applicability of the approach.
>
> **Weaknesses:**
>
> Applicability to alternate sensors:
>
> The simulation was intended to be notably different from the real sensor (depth-based vs marker-based). This had two advantages 1) it allowed for a simple method of simulation and 2) it demonstrated that the domain adaption approach could be used to bridge a large sim2real gap in the tactile domain. We believe that this increases its viability across a range of tactile sensors. We also believe that the type of simulated tactile images most closely relate to optical tactile sensors, including the Soft-Bubble sensor (Kuppuswamy et al 2020), those of Gelsight-type, and marker-based optical tactile sensors such as the TacTip used here.
>
> In principle, we see no reason why the proposed data-driven domain adaption approach would not work for electromechanical tactile sensors such as the BioTac. That said, there are key differences from optical tactile sensors – for example, the BioTac is multimodal with 19 electrodes giving its spatial resolution, whereas optical tactile sensors are typically unimodal with spatial resolution over thousands to millions of pixels, much of which is likely redundant for many tasks. To not overclaim, we have limited statements made in the work to optical tactile sensors, which we believe is justified by the difference between simulated and real tactile data. We hope that our work would also transfer to sensors such as the BioTac, which labs with access to that technology could explore.
>
> Complexities of Data Collection:
>
> We have added some information to section 5.1 to clarify how difficult the data collection stage was. We use a static environment comprised of 3d printed components to represent the tactile stimulus a sensor is likely to receive when evaluated in an environment. The 3d printed components can also be easily loaded into simulation for collecting the corresponding simulated images. For each task, the environment can be broken down into relatively simple stimuli such as flat surfaces, straight edges and spherical probes. As noted in the paper, care is required to ensure shear invariance and was necessary for strong evaluation performance. In terms of time taken for data collection: on the real robot, collection of 5000 tactile images for training took between 3 and 5 hours dependent on the procedure; in simulation this took <100 seconds.
>
> Demonstrating sim-to-real generalization:
>
> We have clarified this further in the text. For most tasks, we do find some generalization:
>
> - Edge following is trained in an environment with straight edges and right-angle corners but when evaluated can navigate rounded edges and sharp internal/external angles both in sim and real.
>
> - Surface following shows generalisation across surface texture and frictional properties that are present in reality but not modelled in sim. The undulating stimulus and 3d-printed stimuli have quite different frictional properties, which will affect sensor shear.
>
> - Object rolling is trained across bearings of size 5mm to 10mm at varying depths. This is shown to extend to a range of 2mm to 8mm during evaluation both on real hardware and in sim.
>
> - Object pushing is trained only with a cube object but evaluated with several different shapes. This shows some generalisation across rounded vs flat contacts, small changes in mass and varying frictional properties. As noted in the paper this was the most sensitive environment.
>
> One of the reasons for choosing end-to-end learning via RL as a pipeline in this project is for its ability to scale and generalise. Future work could make full use of large-scale data available in simulation, and in principle learn far more general policies.
>
> **Issues:**
>
> Oscillations and potential solutions:
>
> As this problem is common across many applications of RL (Bohez et al. [32]) and was not a major issue for the present work, we view it as an orthogonal direction of research and not necessary to explore within the scope of the present work.

---

### Official Review · Reviewer_7RPs · 2021-07-26

**Originality:** Good
**Technical Quality:** Very Good
**Clarity Of Presentation:** Very Good
**Impact:** 3

**Recommendation:**

Weak Accept: I recommend accepting the paper, but will not argue for my recommendation if the majority of other reviewers have a different opinion.

**Summary:**

This paper proposes an efficient method to simulate tactile images and show that RL policies trained within the simulation can transfer to real-world settings with an learned image-to-image translation module that approximately turns real-world image into simulated image. The results are demonstrated on several tasks defined by the authors.


**Issues:**

- May authors clarify what does blue represent in Figure 6? It’s not shown in the legend.


**Reviewer Expertise:**

Good: General knowledge of the area

**Strengths And Weaknesses:**

**Strengths**:
- The paper is well-written and organized. Each section has very clear paragraphs that summarize both the motivations and design decisions.
- The sim-to-real results are encouraging. As collecting tactile images can be very time-consuming in the real world, successful sim-to-real results can encourage more people to explore this direction.
- The released simulation and benchmark can be very useful for the community.

**Weaknesses**:
- (Nitpicking) no major novelty is proposed. Image-to-image translation for sim2real has been explored in more complicated domains (e.g., vision), so showing that the same technique works for tactile is generally less surprising.


**Summary Of Recommendation:**

I recommend accepting the paper because to the best of my knowledge, this is the first paper that presents results on successful sim-to-real tactile-based policy with *domain adaptation*.  Additionally, the paper is well-written and the experiments show real-world results.

---

> ### Author Response · Authors · 2021-08-27
> **Response to Reviewer-7RPs**
>
> We thank the reviewer for their time spent reviewing and constructive comments.
>
> **Strengths:**
>
> Thank you for recognizing these contributions of our work.
>
> **Weaknesses:**
>
> As discussed in response to the AC, there is novelty in modifying an approach from the computer vision community to apply to tasks using tactile sensing, as we appreciate you recognize. For a full response on novelty, we refer to the answer we gave the AC above on ‘Originality’
>
> In our view, there is novelty in the full approach that includes simulating a tactile sensor, creating a suite of tasks to demonstrate performance, performing matching dataset collection, validating that the approach works and highlighting areas where tactile data may have advantages when compared with vision. We hope that this will be of value to the community.
>
> **Issues:**
>
> Clarification on Blue in Figure 6: The blue-red gradient is just used to help visualize the 3d shape of the spherical surface. The error bars indicate the colour of each quiver plot which corresponds to the amount of error between the ground truth and sensor position/orientation. For clarity, we have amended the figure to have a colourless representation of 3d shape.

---

### Official Review · Reviewer_3URa · 2021-07-26

**Originality:** Very Good
**Technical Quality:** Very Good
**Clarity Of Presentation:** Good
**Impact:** 4

**Recommendation:**

Strong Accept: I recommend accepting the paper and will argue for my recommendation even if other reviewers hold a different opinion.

**Summary:**

The paper proposes an approach for simulating optical tactile sensors using depth imaging and approximation with rigid body dynamics. Subsequently, real2sim image translation is utilized to enable policy transfer from simulation to reality. The approach is validated on a suite of tactile RL environments such as edge/surface following and object rolling/pushing/balancing.

**Issues:**

See weaknesses above

**Reviewer Expertise:**

Very good: Comprehensive knowledge of the area

**Strengths And Weaknesses:**

Strengths: the paper proposes a novel and practical approach to bridging the sim2real gap when using optical tactile sensors. The idea of using image translation appears straightforward and useful. The evaluation is thorough, and the source code is provided. The proposed method can be applied to a whole family of optical tactile sensors, making it interesting to a broader community. The suit of tasks proposed in the paper can serve as a basis for future tactile control benchmarks.

Weaknesses: writing can be improved
i) Provide a succinct and self-contained description of the complete approach, e.g., via a diagram or an algorithm box. One can reconstruct roughly what the algorithm is but it would help the reader to see it at once and formulated in a concrete manner.
ii) In general, sections seem hanging independently and there is a lack of cohesive narrative in the paper. It reads a bit like a lab report. Adding some connecting sentences and an overarching narrative would improve the flow.

**Summary Of Recommendation:**

The idea of simulating tactile features using depth images and rigid body dynamics with subsequent image translation is proposed and evaluated. Tactile non-prehensile manipulation tasks are used to evaluate the method. The idea is good and the evaluation is well-executed. The results suggest that the method is viable. Limitations and directions for future work are discussed in detail. This paper may serve as the basis for further investigations into simulation and control based on optical tactile sensors.

---

> ### Author Response · Authors · 2021-08-27
> **Response to Reviewer-3URa**
>
> We thank the reviewer for their advice and endorsement of the paper.
>
> **Strengths:**
>
> Thank you - we agree that the approach is both novel and practical. Also, we intend that the proposed tasks could serve as a benchmark for tactile robotics in general and would be happy to share code/CAD/other materials to help facilitate other groups use this as a platform for comparison.
>
> **Weaknesses:**
>
> In response to point “i) Provide a succinct and self-contained description of the complete approach, e.g., via a diagram” we have re-made Figure 1 to give an overview of the full approach. We hope that this will give the reader an initial overview and therefore improve clarity across the paper.
>
> In response to “ii) In general, sections seem hanging independently and there is a lack of cohesive narrative in the paper,” we have made additional effort to improve readability throughout the work, in particular for section 5 as encouraged by reviewer 3Knc, and to the related work section encouraged by the AC. Due to us being at the limits of allowed space, and that most reviewers found the paper generally clear, changes outside of these sections have been relatively minor.

---

### Meta-Review · Area_Chair_7G38 · 2021-08-16

**Recommendation:** Accept (Poster)
**Confidence:** 5

**Metareview:**

This paper focuses on the problem of sim-to-real for tactile sensing.

Clarity: The paper is generally clear as mentioned by all the reviewers, but as pointed out by Reviewer 3URa the paper could benefit from a bit more polishing. In particular, when it comes to Sec. 5, and how it fits in the general scheme (Reviewer 3URa and 3knc).

Quality: The real-robot experiments are certainly well appreciated by the reviewers, as well as the promise to open-source the code. The work overall has several interesting bits, but the manuscript seems to claim a lot of contributions while failing to thoroughly investigate any of them. To me, it would seem that the main (and more valuable) contribution could be to show how the Real-to-Sim Tactile Image Translation provides superior performance for sim2real transfer. However, the current manuscript does not provide any experiment that compares trivial sim2real to the proposed approach. As such, it is impossible to understand how much of the sim2real performance of Real-to-Sim Tactile Image Translation is due to the simulator being accurate vs the use of the proposed approach. (The simulator itself is also listed as a contribution, but is not evaluated in any way)
In my opinion, adding this experiment and being a bit more precise regarding the actual contributions would improve the manuscript.

Originality: I find the novelty of the paper rather limited. As mentioned in the related work there are several tactile sensing simulators already existing, and the idea of sim-to-real for tactile sensing has also been previously investigated. The paper does not compare to any of them. The algorithm proposed is on itself also adapted with minor changes from the computer vision community, as pointed out by Reviewer 7RPs.

Significance: Sim2real for tactile sensing is certainly an important and timely topic. While this is not the first paper dealing with the problem, a strong demonstration that the proposed method improved over trivial sim2real and/or existing methods would be valuable to the community.

Other comments:
- Related works do not currently provide an accurate depiction of the existing literature in tactile sensing simulators and sim2real, and as such should be improved. Related works in sim2real for tactile sensing should be explicitly discussed (some are already cited in other parts of the paper).
- In the experiments of Fig.3 the fact that the Oracle does not always provide the best performance is troubling, and I am not convinced by the explanation of Sec. 4.3. To me, this seems to indicate either that the Oracle is not providing sufficient information (hence it shouldn't be called Oracle in the first place), or that the learning algorithms are not properly tuned.

---

> ### Author Response · Authors · 2021-08-18
> **Clarification on trivial sim2real approach**
>
> We thank the AC for taking time to go through the paper and give further useful feedback in addition to the 4 constructive reviews. If possible, we would like some clarification on one of your points.
>
> Can more detail be given on what exactly is the "trivial sim2real” approach suggested for use in an “experiment that compares trivial sim2real to the proposed approach”? From our understanding this is asking for the policies learned via simulation to be directly applied to the physical setup without any image modification, but we are not sure if we have understood you correctly.

---

> > ### Comment · Area_Chair_7G38 · 2021-08-18
> > **Clarification on trivial sim2real approach**
> >
> > Sorry for the ambiguity, yes with "trivial sim2real" I meant to directly apply the policies learned in simulations to the real setting. This would provide a baseline that can be used to compare the benefits of using the proposed approach.
> >
> > Evaluating one or more of the existing methods in the literature would also provide valuable baselines.

---

> > > ### Author Response · Authors · 2021-08-19
> > > **Clarification on trivial sim2real approach**
> > >
> > > Thanks for the clarification. We are happy to do these new experiments, but suspect their performance will be too poor for a useful baseline – e.g. fig 1 shows that the simulated and real tactile images look very distinct. This was intentional to keep the simulation agnostic to the type of tactile sensor by focusing on depth features (which are non-trivially related to e.g. marker shear).
> > >
> > > We take your point on finding other baselines. However, we are struggling to find suitable comparisons as to the best of our knowledge published methods either use simulations (e.g. FE) incompatible with sample inefficient methods such as RL or rely on specific hardware (e.g. BioTac, GelSight). Please let us know if you are aware of an appropriate baseline though.
> > >
> > > If we’re not able to find a suitable baseline, a way forward would be to explain this in the paper, which would fit with your comment that ‘related work in sim2real for tactile sensing should be explicitly discussed’.

---

> ### Author Response · Authors · 2021-08-27
> **Response to AC-7G38 (part 1)**
>
> We thank the AC and reviewers for giving constructive and insightful feedback.
>
> **Clarity:**
>
> In order to improve clarity of the overall method, Figure 1 has been modified to include an overview of the full approach. This is in response to points raised by reviewer 3URa “Provide a succinct and self-contained description of the complete approach, e.g., via a diagram”. In addition to this, Section 5 has been extended to include more detail on “how complicated is the setup for such data collection” as posed by reviewer 3Knc. We also add detail on how we use 3d-printed shapes to approximate a task and the time taken to collect this data in both simulation and reality.
>
> **Quality:**
>
> We thank the reviewers and AC for viewing real robot experiments and open-sourcing of code as valuable.
>
> In response to some perceived overclaiming of contributions, we have aimed to add more support for our claims where possible as we describe below. Additionally, throughout the paper we have aimed to be more precise regarding our contributions.
>
> Regarding the comparison with trivial sim2real:
>
> We believe that a comparison to trivial sim2real (i.e. direct policy transfer) will not serve as a useful baseline as the simulated sensor is significantly different from the real sensor used in this work. Therefore, the directly transferred policies will not behave as intended. As previously commented, this decision was intentional to keep the simulated sensor agnostic to the type of real tactile sensor used. We have now more explicitly stated this motivation in the paper, both in section 2 (“Hence, we keep the simulation agnostic to the sensor transduction, which should facilitate future application of these methods to other types of tactile sensor.”) and section 3 (“we do not attempt to simulate images accurate to any specific tactile sensor...”).
>
> To check this intuition, we performed these experiments and confirmed that the policies were poor. Descriptions of behavior are as follows:
> - Edge Following: The actions deterministically led the sensor off the edge towards the safety limits.
>
> - Object Rolling: As the goal location is included in the observation, the actions would start by moving in the general direction of a goal. The goal was however not successfully reached and the object rolled off the sensor. This resulted in 0/25 successful trials compared with 25/25 when using translated images.
>
> - Object Pushing: As both the TCP position and goal position are included in the observation the robot would initially move the object correctly. However, the object would soon rotate around the sensor and break from the desired trajectory.
>
> - Surface Following: We deemed this too unsafe to test as incorrect actions are likely to damage the sensor.
>
> In response to further comments regarding improved comparison to existing methods. It seems that part of the issue, as noted by the AC, was due to a lack of “accurate depiction of the existing literature in tactile sensing simulators and sim2real”. In response to this we have re-written the entire Related Work section to better position the work amongst existing literature. There are advantages to the proposed approach that we have now highlighted, including simulation speed allowing for use with more general sample-inefficient methods like DRL and increased applicability to alternative tactile sensors due to the data-driven domain adaption approach.
>
> We were not able to find an existing sim-to-real tactile reinforcement learning baseline to compare to that did not require specific hardware or simulation methods that did not straightforwardly apply to our reinforcement learning tasks (in that the simulations would be too slow or require specific hardware we do not have access to).
>
> Instead, for a baseline, we provide an additional experiment comparing to previous work simulating the same type of optical tactile sensor ([11] Ding et al, ICRA 2020). We have performed a direct comparison by recreating three supervised learning tasks, which are detailed in the additional Appendix G. As now included in the paper: “we find a 3-fold improvement in Mean Absolute Error (MAE) when predicting radial displacement and polar angle of an edge pressed into the sensor”. We do not report on results from Task III in the main body of text as there is a difference between task setup; we use a flat tip to make better use of data, GANs and 3d printed objects previously required for the object rolling task, whereas previous work used a hemispherical tip.
>
> Table 1: Sim2Real Mean Absolute Error evaluated on three supervised learning tasks.
>
> | Approach | Task I | Task II | *Task III |
> | ----------- | ----------- | ----------- | ----------- |
> | Ours      | **0.079**   | **0.119**  | **0.059** |
> | Ding et al. | 0.254      | 0.45     | 0.73 |
>
> Due to space limitations, the details of this analysis have been included in Appendix G with the results summarized in the paper (Section 5.3).

---

> > ### Author Response · Authors · 2021-08-27
> > **Response to AC-7G38 (part 2)**
> >
> > **Originality:**
> >
> > The majority of reviewers define the work as novel or list a lack of “major” novelty as a mild criticism (Reviewer 3URa: “proposes a novel and practical approach to bridging the sim2real gap”, Reviewer 3J9a: “It is novel to apply GAN to bridge the real2sim gap for tactile signals... extra consideration to disentangle the influence of shear force for better real2sim.”, Reviewer 7RPs: “(Nitpicking) no major novelty is proposed.”, “first paper that presents results on successful sim-to-real tactile-based policy with domain adaptation.”).
> >
> > Overall, there is novelty in modifying an approach from the computer vision community to apply to tasks using tactile sensing. It was by no means a straightforward application of taking an existing method to apply to a different domain, because touch behaves so differently from vision. There were a number of challenges that needed addressing, and so we emphasize the components required to successfully apply the approach to touch, including: 1) separate static dataset collection in a safe and controlled manner 2) taking care to disentangle shear; 3) reduction of image artefacts through algorithm changes; and 4) border removal/addition to achieve successful GAN training.
> >
> > There is also novelty in creating and open-sourcing a platform that allows for end-to-end learning of policies from high-resolution tactile data, then demonstrating real-world successful policy transfer.
> >
> > **Significance:**
> >
> > We agree that “Sim2real for tactile sensing is certainly an important and timely topic” and intend that this work will enable other work in the field of robot dexterity, such as with other optical tactile sensors and on other challenging tasks (such as the balancing task we propose as a challenge for the field).
> >
> > **Other comments:**
> >
> > Related Work: As noted above we have redone the related work to better position the paper amongst existing literature.
> >
> > Oracle Observation: In response to this comment, we have changed the name from `Oracle’ to ‘Env. State’ to not be misleading. The full details of Env. State information are given in Appendix C. As the Env. State observation is faster to run (no image rendering, smaller NNs), the majority of the tuning was performed in this domain. However, no explicit hyper-parameter tuning was performed and therefore we cannot confidently say that the algorithms were tuned properly to Env. State observations, especially given that RL algorithms are notoriously fragile. Thus, while we believe the explanation given in Sec 4.3 is correct, we have removed it as it was not explicitly verified.

---

### Decision · Program_Chairs · 2021-09-13

**Decision:**

Accept (Poster)

**Comment:**

This paper focuses on the problem of sim-to-real for tactile sensing.

Clarity: The paper is generally clear as mentioned by all the reviewers, but as pointed out by Reviewer 3URa the paper could benefit from a bit more polishing. In particular, when it comes to Sec. 5, and how it fits in the general scheme (Reviewer 3URa and 3knc).

Quality: The real-robot experiments are certainly well appreciated by the reviewers, as well as the promise to open-source the code. The work overall has several interesting bits, but the manuscript seems to claim a lot of contributions while failing to thoroughly investigate any of them. To me, it would seem that the main (and more valuable) contribution could be to show how the Real-to-Sim Tactile Image Translation provides superior performance for sim2real transfer. However, the current manuscript does not provide any experiment that compares trivial sim2real to the proposed approach. As such, it is impossible to understand how much of the sim2real performance of Real-to-Sim Tactile Image Translation is due to the simulator being accurate vs the use of the proposed approach. (The simulator itself is also listed as a contribution, but is not evaluated in any way)
In my opinion, adding this experiment and being a bit more precise regarding the actual contributions would improve the manuscript.

Originality: I find the novelty of the paper rather limited. As mentioned in the related work there are several tactile sensing simulators already existing, and the idea of sim-to-real for tactile sensing has also been previously investigated. The paper does not compare to any of them. The algorithm proposed is on itself also adapted with minor changes from the computer vision community, as pointed out by Reviewer 7RPs.

Significance: Sim2real for tactile sensing is certainly an important and timely topic. While this is not the first paper dealing with the problem, a strong demonstration that the proposed method improved over trivial sim2real and/or existing methods would be valuable to the community.

Other comments:
- Related works do not currently provide an accurate depiction of the existing literature in tactile sensing simulators and sim2real, and as such should be improved. Related works in sim2real for tactile sensing should be explicitly discussed (some are already cited in other parts of the paper).
- In the experiments of Fig.3 the fact that the Oracle does not always provide the best performance is troubling, and I am not convinced by the explanation of Sec. 4.3. To me, this seems to indicate either that the Oracle is not providing sufficient information (hence it shouldn't be called Oracle in the first place), or that the learning algorithms are not properly tuned.